# Comparison of Endodontic Failures between Nonsurgical Retreatment and Endodontic Surgery: Systematic Review and Meta-Analysis with Trial Sequential Analysis

**DOI:** 10.3390/medicina58070894

**Published:** 2022-07-04

**Authors:** Mario Dioguardi, Camilla Stellacci, Lucia La Femina, Francesca Spirito, Diego Sovereto, Enrica Laneve, Massimo Francesco Manfredonia, Alfonso D’Alessandro, Andrea Ballini, Stefania Cantore, Lorenzo Lo Muzio, Giuseppe Troiano

**Affiliations:** 1Department of Clinical and Experimental Medicine, University of Foggia, Via Rovelli 50, 71122 Foggia, Italy; camilla_stellacci.546887@unifg.it (C.S.); lucia_lafemina.560051@unifg.it (L.L.F.); spirito.francesca97@gmail.com (F.S.); diego_sovereto.546709@unifg.it (D.S.); enrica.laneve@unifg.it (E.L.); drfrancomanfredonia@gmail.com (M.F.M.); alfonso.dalessandro@unifg.it (A.D.); lorenzo.lomuzio@unifg.it (L.L.M.); giuseppe.troiano@unifg.it (G.T.); 2Department of Precision Medicine, University of Campania “Luigi Vanvitelli”, 80138 Naples, Italy; andrea.ballini@me.com; 3Independent Researcher, 70129 Bari, Italy; stefaniacantore@pec.omceo.bari.it

**Keywords:** endodontic, endodontic retreatment, apicoectomy, surgical endodontic retreatment, trial sequential analysis, endodontic failure, endodontically, endodontic surgery

## Abstract

*Background and Objectives*: In the presence of a persistent endodontic lesion or endodontic failure, the alternative for the recovery of the dental element is endodontic retreatment or endodontic surgery, which consists in the surgical removal of the root apices with retrograde closure of the endodontium. The objective of this systematic review and meta-analysis was to provide an updated value of the Risk Ratio between the two types of treatment in order to offer to clinicians who propose a non-surgical endodontic retreatment or an endodontic surgery a direct comparison. *Materials and Methods:* The revision was performed according to PRISMA indications: three databases (PubMed, Scopus and Cochrane register) were consulted through the use of keywords relevant to the revision topic: surgical endodontic retreatment, endodontic retreatment, apicoectomy. This search produced 7568 records which, after eliminating duplicates and applying the inclusion and exclusion criteria, resulted in a total of seven included articles. The meta-analyses were conducted by applying fixed-effects models, given the low percentage of heterogeneity. In addition, trial sequency analysis (TSA) was performed for the analysis of the statistical power of the results and GRADE for the quality of the evidence. *Results:* The results of the meta-analyses’ data report an aggregate risk ratio (RR) between non-surgical endodontic retreatment and surgical endodontic retreatment of: 1.05 [0.74, 1.47] at one year of follow-up; RR 2.22 [1.45, 3.41] at two years of follow-up; an RR 1.08 [0.73 1.62] for a follow-up period of 3–4 years; and an RR 0.92 [0.53, 1.61] for a follow-up period of 8–10 years. *Conclusions:* The results of the present meta-analysis show that in the long term, the risk of failure is identical for the two groups, and there is only a slightly higher risk of failure for non-surgical endodontic retreatments, when only two years of follow-up are considered.

## 1. Introduction

The primary purpose of endodontic treatment is the complete disinfection and cleansing of endodontic canals from the presence of microorganisms and from potentially infected pulp endodontic tissue, with the aim of obtaining a completely shaped and disinfected space that can accommodate the canal filling material and obtain closure three-dimensional endodontium with a reduction in the possibility of recurrence of the endodontic pathology.

The primary treatment of the canal does not always achieve its objectives, and after years, there may be endodontic failure, the prevalence of which could reach about 30% of endodontically treated teeth, as reported in large cross-sectional studies [1].

The causes of a failure are to be found in the persistence of bacteria inside the endodontium, due to the presence of spaces not adequately left empty and/or cleaned and disinfected. in this way, the bacteria have the possibility, if in contact with the external biological fluids, to survive and proliferate giving a secondary lesion [2]. Moreover, bacteria such as *Enterococcus faecalis*, resistant to the action of irrigants, especially if not used for an appropriate time, can survive in the dentinal tubules, aggregating itself in structures such as the biofilm [3].

The biofilm can also consist in *Actinomycetes* spp. (*Propionibacterium* and *Actinomyces*), affecting the dental element external root surface and leading in this way to a persistent extra root lesion [4].

In the presence of a persistent endodontic lesion or endodontic failure, the options for the recovery of the dental element are non-surgical endodontic retreatment or surgical endodontic retreatment, which consists in the surgical removal of the root apices with retrograde closure of the endodontium [5,6].

In accordance with Karabucak and Setzer [7], the criteria which should guide the clinician in choosing between performing a non-surgical endodontic retreatment and a surgical endodontic retreatment concern the following points:I.Evaluation of the coronal seal: the presence of an adequate coronal restoration or a still functional prosthetic crown that seals the endodontium coronally or whose removal involves the non-restorability can lead to the choice towards surgical endodontic retreatment. On the contrary, a restoration no longer adequate, with good access to the endodontium, may lead to the choice of non-surgical endodontic retreatment [8].II.Radiographic evaluation of root canal obturation: the presence of untreated canals and a coarse filling of the canals may lead to the choice of non-surgical endodontic retreatment. On the contrary, a surgical approach could be recommended in the presence of an apparent adequate apical seal or in the presence of a root canal obturation that is far from the radiographic apex of only 1 mm (in the presence of course of symptoms of a periapical pathology), and a surgical approach should be evaluated [9,10,11].III.The presence of clinical complications such as the finding of fractured instruments (apical third of the root), the presence of zipping of internal resorptions and the presence of root perforations or ledge formations; in these cases, the surgical approach may be a suitable choice [12].IV.In the presence of recurrent infections in which a root canal retreatment has already been carried out, a surgical endodontic retreatment is absolutely to be considered as a therapeutic choice [13].

The survival rate of teeth that received non-surgical endodontic retreatment was 85% after 72 months, 86.8% after 48 months and 90% after 24 months. The survival rate of teeth that received surgical endodontic retreatment was 88% after 72 months, 90.5% after 48 months and 93.7% after 24 months [14]. These data are also in agreement with the most recent systematic reviews conducted by Alghamdi et al. [15] and by Pinto et al. [16]. In particular, the latter reports a success rate of 91.3% for randomized controlled trials (RCTs) and 78.4% for prospective controlled trials (RCTs) PCTs with a very wide follow-up range from 2 to 13 years.

There is only a previous systematic review conducted by Del Fabbro et al. in 2010 [17], which directly compared the success/failure rates of non-surgical endodontic retreatment and surgical endodontic retreatment. In particular, Del Fabbro et al. included only two studies [17], for instance, Danin et al. [18] and Kvist and Reit [19], and the results show that the RR (Risk Ratio) at a one-year follow-up period was 1.13 [0.98, 1.30], slightly in favor for surgical endodontic retreatment. For the other follow-up periods, the results of the meta-analysis carried out concern only the single studies and therefore are insufficient for meta-analytical conclusions.

However, two other reviews by Torabinejad et al. [5] and Kang et al. [20] indirectly compared the success rates of non-surgical endodontic retreatments with surgical endodontic retreatment, including many more studies, but the comparison between the two treatments was lacking in the individual studies included.

Taking into account these premises, the aim of this systematic review and meta-analysis was to provide, in the light of new studies and the advent of new endodontic biomaterials such as bioceramics [21,22,23], an updated RR value between the two types of treatments (non-surgical endodontic retreatment and surgical endodontic retreatment) in order to provide the clinicians with a direct comparison of the successes and failures between the two methods.

## 2. Materials and Methods

### 2.1. Protocol and Registration

The writing of the review was carried out following the indications of the PRISMA (Preferred Reporting Items for Systematic Reviews and Meta-Analysis) and was conducted based on the indications of the Cochrane handbook [24,25]; the protocol, search strategy, inclusion criteria, and search outcomes were recorded and specified prior to the screening and search of articles on PROSPERO (the International database of prospectively registered systematic reviews) with registration number CRD42021273533.

### 2.2. Eligibility Criteria

All retrospective prospective studies and RCTs evaluating the number of failures of teeth undergoing non-surgical endodontic retreatment and those undergoing surgical endodontic retreatment with retrograde filling were considered. In addition, the PICO question was formulated, and the following points were identified: (P)articipants (patients with dental elements whose endodontic therapy has failed), (I)ntervention (teeth with failure of endodontic therapy undergoing non-surgical endodontic retreatment), (C)ontrol (teeth with failure of endodontic therapy undergoing surgical endodontic retreatment) and (O)utcome (Risk ratio between teeth undergoing non-surgical endodontic retreatment and teeth undergoing surgical endodontic retreatment). The PICO question, therefore, was the following: “What is the risk ratio between teeth undergoing non-surgical endodontic retreatment and those undergoing surgical endodontic retreatment?”.

The inclusion criteria were: (1) studies that clearly indicated the number of teeth undergoing non-surgical endodontic retreatment and (2) the number of teeth undergoing surgical endodontic retreatment with related failures. The studies to be included must necessarily have investigated both methods of retreatment in the same patient court.

The exclusion criteria were: (1) studies published in a language other than English, (2) those that did not present data on retreatment failures and successes and (3) those at high risk of bias.

Exclusion criteria were not considered studies for surgical and endodontic techniques that involved the use of microscopy, ultrasonic instruments and biomaterials such as bioceramics.

### 2.3. Sources of Information, Research and Selection

The studies were identified through bibliographic searches in electronic databases by two investigators (M.D. and D.S.). Restrictions on the language of publication have been applied, and articles in a language other than English have been excluded. The search was performed on 3 different databases: Scopus, PubMed, and Cochrane library registry. The latest literature search was conducted on 11 March 2022. In addition, Google Scholar and Open Gray literature search were also consulted, for sources not otherwise identifiable, and systematic reviews were investigated in search of further records.

We used the following database search terms: surgical endodontic retreatment, endodontic retreatment, apicoectomy. Duplicate results were removed using the EndNote x8 software (Thomson Reuters, New York, NY, USA), and the overlaps of articles that could not be uploaded to EndNote were manually removed after the screening phase. The records identified were assessed and examined by 2 reviewers (MD and DS) separately, the screening included the analysis of the title and abstract, and the studies potentially eligible to be included in the systematic review were read full text; in addition, a third reviewer (A.B.) was tasked with resolving doubtful situations.

### 2.4. Data Collection Process and Data Characteristics

The data to be extracted were previously established by the 2 authors responsible for screening the articles and were reported independently in 2 tables to be compared later to reduce the risk of errors. The data extracted from the studies concerned the first author, the type of study, the year of publication, the country that conducted the study, the number of teeth, the number of patients, the endodontic filling used, the type of retreatment performed, the causes of the retreatment failure, the number of failures and the follow-up period.

### 2.5. Risk of Bias in Individual Studies, Summary Measures, Summary of Results, Risk of Bias between Studies and Additional Measures

The risk of bias in the individual studies was assessed by one author (M.D.), with a second author in charge of verifying the correct assessment (D.S.) For the risk of bias, a different scale was adopted from that proposed by the Cochrane handbook, because the latter is studied for RCTS, as the current systematic review, also includes prospective or retrospective non-randomize studies. It was considered to adopt a different tool evaluation checklist used by Mcfarlane (2001) [26,27]. Studies evaluated with a high risk of bias were excluded from the meta-analysis. The results were summarized and graphically represented using the Forest Plot with indications of the inconsistency indices such as the Higgins index (*I*^2^).

The risk of bias between studies was evaluated graphically through the analysis of the overlaps of the confidence intervals, through the *I*^2^. An *I*^2^ value greater than 75% was considered high, and an analysis of the effects was applied, in specific cases and through a funnel plot: the meta-analysis presented high heterogeneity indices, and a sensitivity analysis was performed, excluding only studies that had low confidence interval overlap or that emerged graphically from the funnel plot.

For the meta-analysis, and in particular for the calculation of the aggregate RR, the software Reviewer Manager 5.4 (Cochrane Collaboration, Copenhagen, Denmark) was used. We used the online software GRADE pro-Guideline Development Tool (GRADE pro-GDT, Evidence Prime, Hamilton, ON, Canada), to assess the quality of the evidence. The trial sequency analysis (TSA) was performed using Stata 13 (StataCorp, College Station, TX, USA), with the implementation of the R 4.2 software and by installing the idbounds and metacumbounds commands [28].

## 3. Results

### 3.1. Selection of Studies

The search in the Cochrane library databases, Scopus and PubMed provided in total 7568 bibliographic citations, removing the overlaps manually and using software (Endnote) and excluding the articles that did not meet the legibility criteria when reading the abstract, resulting in 54 potentially admissible articles. Following the full text reading, the number was reduced to only 7 studies that met the inclusion and exclusion criteria and were included in the meta-analysis. Furthermore, the gray literature analysis (Google Scholar, Open Gray) and the previous systematic reviews did not allowed to identify additional studies to be included in the meta-analysis (Figure 1). Finally, on 16 March 2022, an update of the research on Scopus and PubMed was carried out: all keywords and record search details are also represented in Table 1.

### 3.2. Data Characteristics

Articles included in the meta-analysis are as follows: Danin et al. (1996) [18], Curtis et al. (2018) [29], Calişkan (2005) [30], Kvist and Reit (1999) [19], Allen et al. (1989) [31], Riis et al. (2018) [32] and Ercan et al. (2007) [33].

Only 7 studies were included in the meta-analysis; the articles included Retrospective Clinical Studies and RCTs, whose data were published between 1989 and 2018, and included a follow-up period from 1 to 10 years. The total number of patients investigated was 1833 (including those lost during follow-up and those excluded from retreatment), with 579 teeth retreated non-surgically and 454 retreated surgically, and most of the studies performed an orthograde and retrograde filling with gutta-percha: Curtis et al. (2018) [29] reports MTA as retrograde closure, while Danin et al. (1996) [18] glass ionomer cement. Furthermore, the main cause of failure was the presence of endodontic lesions and vertical root fracture. All data relating to the number and causes of failure were extracted and reduced in Table 2.

Data on treatment failure have been grouped by years of follow-up, in order to simplify the execution of meta-analysis; the periods considered were 1 year, 2 years, 3–4 years and 8–10 years of follow-up.

### 3.3. Risk of Bias

Risk of bias was assessed using the checklist used by Mcfarlane (2001) [26], which modifies 2 checklists created by Downs and Black [34] and Harvey [35] for epidemiological, cohort, cross sectional and case control studies, and modified by the authors to adapt it to studies in dentistry, as already done in previous systematic reviews with meta-analyses. The results were reported in Table 3.

To each category was assigned a value from one to 5 (where one = low and five = high). The questions that the review answered by assigning the score were the following:(1)Non-response rate: Is the participation on/follow-up rate stated? Do the authors describe the effort to increase the participant/follow-up rate?(2)Representativeness of sample to target population: Were the subjects asked to participate in the study representative of the entire population from which they were recruited?(3)Validity and reliability of outcome measurement: Were the main outcome measures used accurate (valid and reliable)?(4)Amount of loss to follow-up: Are the non-participants/subjects lost to follow-up described? Do the authors describe the effort to increase the participation/follow-up rate?(5)Appropriate statistical tests: Are the statistical methods described?

Studies and reports presenting a high risk of bias were not included in the analysis quantitative. Articles with a high risk of bias were excluded from the table and eliminated during the inclusion phase. The assessment of the risk of bias of the 7 included studies was conducted by the first Author (M.D.).

### 3.4. Metanalysis

The meta-analysis was performed using the Rev manager 5.4 software: in total 4 meta-analyses were performed by calculating the aggregated RR as a function of the follow-up periods of 1-year, 2-years, 3–4-years and 8–10-years follow-up, in order to reduce the bias between the data in the different studies and consequently to reduce the bias and heterogeneity between the studies. The aggregate RR was calculated for each analysis by applying a fixed effects model.

For the calculation of the aggregate RR for a one-year follow-up period, we have data from only 3 studies: Danin et al. (1996) [18], Allen et al. (1989) [31] and Ercan et al. (2007) [33]. A fixed effects model was applied; the heterogeneity is low with Chi^2^ = 3.40, df = 2 (*p* = 0.18), and the Higgins index (*I*^2^) is 41%.

The results of the first meta-analysis show an aggregate RR of 1.05, with the related confidence intervals [0.74 1.47]. The test for the overall effect is Z = 0.26 (*p* = 0.80). The forest plot shows a neutral RR between the 2 groups (non-surgical endodontic retreatments and surgical endodontic retreatments). All the three studies intercept the line representing the confidence intervals, the central line of no effect (Figure 2).

The second meta-analysis concerns the failure data for a 2-year follow-up period. The included studies were 4: Ercan et al. (2007) [33], Kvist and Reit (1999) [19], Calişkan (2005) [30] and Curtis et al. (2018) [29]. The 2-year results are in favor for surgical endodontic retreatments, showing an aggregate RR of 2.22 [1.45 3.41]; the heterogeneity is low Chi^2^ = 4.94, df = 3 (*p* = 0.18), and *I*^2^ was 39% (Figure 3).

The third meta-analysis covered failure data over a 3–4-year follow-up period, presenting only 3 studies providing data over this time period. The forest plot shows how in the first meta-analysis, in a position of equilibrium between the 2 types of reprocessing, the heterogeneity between the studies is absent, as evidenced by the Higgins index *I*^2^ = 0 (Figure 4).

The last meta-analysis included only 2 studies, which provided for a follow-up period of 8–10 years: also, in this case, we have a neutrality of the risk of failure between surgical versus non-surgical endodontic retreatments (Figure 5). However, the strong limit is given by the only inclusion of 2 studies, with a low number of overall patients included, an aspect that was addressed in the trial sequential analysis subparagraph (TSA).

The risk of bias between the studies for all four meta-analyses data was also assessed through a graphical analysis of the funnel plots, which confirmed the absence of heterogeneity between the different studies included (Figure 6).

### 3.5. Trial Sequential Analysis, Grade

The TSA was performed to evaluate the potency of the result of the first and second meta-analysis, adjusting the results to avoid type I and II errors. The software used was Stata 13 (StataCorp, College Station, TX, USA), with the integration of the R 4.2 software via the Metacumbounds commands as described by Miladinovic et al. [36].

The O’Brien–Fleminge expense function was used by applying fixed effects. The AIS (accrued information size) and subsequently APIS (a priori information size) commands were used by the Dialog BOX to determine the optimal sample size and for the power of the results, assuming a RRR (relative risk reduction) of 30%, an Alpha value equal to 5% (type 1 error) and a beta value at 20% (type 2 error) (Figure 7).

The TSA of the failure data included in the first meta-analysis highlights how the “Z” curve does not cross (Z = 1.96) indicating no evidence (Figure 7B). The meta-analysis included fewer patients than the required information size.

Instead, from the TSA of the failure data included in the second meta-analysis, it was noted how the “Z” curve crosses (Z = 1.96), providing a significant result but a spurious effect because the “Z” curve does not exceed the monitoring limit (Figure 7D).

If in the first TSA, there was no evidence, with the possibility of having a false negative result. As an alternative, in the second TSA, we have a statistically significant result, but with the possibility of a false positive result. Furthermore, from the APIS graph, it is clear how the optimal number of patients to be included considering a RRR of 30% is 2578.

The authors also performed a GRADE pro-GDT, to evaluate the quality of the results (Table 4). The results show outcomes in the first, third and fourth meta-analyses, while the quality was moderate for the second meta-analysis results.

## 4. Discussion

In this work, we conducted a systematic review with meta-analysis of data published in the literature on endodontic retreatments, trying to provide a complete picture of what was the least risk of failure treatment between surgical versus non-surgical endodontic retreatment, providing the most up-to-date data to the endodontist practitioner. In addition, to our knowledge, this is the first meta-analysis with TSA that included only studies that investigated the two therapeutic choices, with a direct comparison.

The previous 2010 systematic review conducted by Del Fabbro et al. [17] shows only two studies included, and the extracted data were divided by follow-up periods of 1 and 2 years: these data, even if obtained with extreme methodological rigor (in fact a Cochrane systematic review was performed), may lack an adequate statistical power given the small number of patients included, as it also emerges from performing the TSA on the data extracted from this meta-analysis. In fact, in this systematic review with meta-analysis, we included 7 studies with a number of 1833 endodontic retreatments.

Analyzing the failures between the various follow-up periods, it emerges that, for the follow-up periods of 1 year, 3–4 years and 8–10 years, there is no systematic difference in the failure rates between surgical endodontic retreatments and non-surgical endodontic retreatments, while for the period a period of 2 years, the meta-analysis reports an aggregate RR of 2.2 in favor of surgical endodontic retreatments. Numerous studies agree that a follow-up period of 2 years is a suitable period of time to obtain valid results on success in retreatments [37]. Indeed, as Calişkan reports in his study, about 80% of all cases healed and 70% of all failed cases were evident within 2 years of treatment [30].

In fact, the TSA conducted on the RR data on failures at one year shows that for the three included studies, there is no adequate statistical power of the results, indicating the absence of evidence, with a possibility of a false negative result; instead, if we consider a follow-up period of 2 years, the condition changes, and the TSA shows statistically significant data, but with the possibility of incurring a false positive result for the presence of a spurious effect determined by confounding factors, indicating how, in order to be confident of a true positive result, the optimal number of the patients included in the studies is 2578.

The majority of the systematic reviews with meta-analyses data, including the Cochrane studies, present a number of studies with randomized participants which is too low to hold the adequate statistical power that would allow an adequate evaluation of the intervention effect size. The results deriving from a meta-analysis with few studies, and consequently with few participants, could be not very credible, with the risk of overestimation or of being underestimated, as they do not possess sufficient statistical power to accept or reject interventions even with large effect sizes [38].

The results shown by the TSA are also confirmed by the evaluation of the quality of evidence (Grade), where moderate certainty is obtained for the aggregate RR for a follow-up period of 2 years.

However, the individual studies agree that there is not a big difference between the failure rates between the two therapeutic choices.

In 2018, Riss et al. [32] reported survival rates of 74% for surgical endodontic retreatments and 77% for non-surgical endodontic retreated teeth over a 10-year follow-up period. The main causes of retreatments were (1) non-healing apical periodontal lesion and (2) dental fracture, according to our extracted data (Table 2).

Although Calişkan et al. [30] reports that in the short term of 2 years the data are slightly in favor of surgical endodontic retreatments, according to Danin et al. (1996), this is valid for a one-year follow-up [18].

The main risk incurred by reading the resulting data of this meta-analysis is that of over-interpreting the data at 2 years: the analysis we conducted did not show statistically significant differences compared to 1, 3–4 and 8–10 years. Suggesting one treatment or the other, the 2-year period could be considered suitable in many studies, but the success of an endodontic therapy should be evaluated in reasonably longer follow-up periods, consistent with the planned duration of an endodontic therapy.

Surgical endodontic retreatment has disadvantages that could nullify the improvement obtained in the short term. In fact, Riis et al. [32] reports as cases of failure of the surgical endodontic retreatment group at 10 years, two cases of vertical root fracture and seven cases of non-healing. The apical seals, in this case, were obtained by using gutta-percha for both groups.

Furthermore, it is increasingly clear from the data in the literature that vertical fracture is among the main causes of failure in teeth with surgical endodontic retreatment. In fact, this issue emerges from a study on the incidence of vertical root fractures between conventional versus surgical endodontic retreatment conducted by Karygianni et al. [39], in which 62.31% of the teeth with vertical root fracture had undergone a combination of conventional endodontic retreatment in addition to surgical endodontic retreatment. Other causes for the persistence of the lesion could be found in the quality of the existing root canal filling or in the solubility of the retrograde filling.

In this review, we also focused on the type of biomaterials used to obtain the apical seal, in order to understand if it could represent a cause of a different failure rate.

According to a recent literature review conducted on root canal filling materials performed by Komabayashi et al. [40], the Tricalcium silicate (MTA/Bioceramic) and the Salicylate (MTA fillapex, sealepex, apexit) are the categories of endodontic sealer with the major bioactive components. In fact, studies on the cytotoxicity of MTA revealed biomineralization and osteo-inductivity capabilities, with the additional data showing that the cytocompatibility characteristics were clearly superior to those of other endodontic sealers [41].

The limitations of this systematic review are that the results, even if in part significant, show a small number of participants included, as also highlighted by the TSA.

The results of this study indicate that clinicians can choose either non-surgical retreatment or root-end surgery after failed primary root canal therapy. Finally, only in the short-term period (2 years), with balanced RR values, data regarding surgical endodontic retreatments seem more in favor in terms of healing.

As shown in the results (Table 2), Curtis et al. [29] perform root-end fillings for surgical endodontic retreatments with gray or white ProRoot MTA (Dentsply, Tulsa, OK, USA) or EndoSequence BC Root Repair Material (Brassler USA, Savannah, GA, USA), while for retreatments, an endodontic sealear based zinc oxide eugenol (Roth 811-Roth international, Chicago, IL, USA) was used.

Ercam et al. [33] performed root canal closure through the use of gutta-percha points and a Salicylate Ca (OH)_2_ (Sealapex; Sybron/Kerr, Romulus, MI, USA): they reported 14 endodontic failures (13 in the non-surgical endodontic retreatment group and 1 in the surgical endodontic retreatment group), of which in total, 10 causes were due to an error in filling the canal (underfilling, overfilling, poorly condensed filling), 2 were due to broken instrumentation and, finally, 2 were due to loss of the coronal seal.

In the other included studies, bioceramics or more generally, Tricalcium silicate or Salycilati, were not used as root-end fillings.

Indeed, the studies by Kvist and Reit [19], Allen et al. [31] and Riss et al. [32] proposed the use of gutta-percha; Danin et al. [18] proposed the use of glass ionomer cement (Chem-fill II, De Trey, Zurich, Switzerland) for apical closure of teeth undergoing surgery, while Calistan [30] performed the canal closures using a silicone type sealer (Diaket-ESPE, Seefeld, Germany), with gutta-percha cones and calcium hydroxide paste being used for the apical closure of the surgically treated teeth. For these studies, the causes of failure were extensively described in Table 2.

The data of the failures extrapolated and presented in Table 2 do not reveal sufficient data to be able to perform an analysis of the subgroups, according to the type of sealer used in the course of surgical endodontic retreatment.

The analysis of possible biases within the present systematic review was addressed and planned on three fundamental points. The first point is the presence of a publication bias; the real issue was the small number of articles included (seven for this systematic review, Table 2). Publication bias is a problem that afflicts systematic reviews with meta-analyses data because the scientific literature tends not to publish statistically insignificant data. An attempt has been made to minimize it through analysis of the gray literature and of the sources that also includes abstracts of conferences, in order to intercept those reports of studies performed but not published [42].

Furthermore, to assess the presence of a probable publication bias, a visual analysis of the symmetry of the funnel plots was performed. However, it was not possible to clearly evaluate the issue due to the small number of studies [43]. Furthermore, in the light of the few included studies, TSA was performed to determine if there was an adequate number of participants and if the results possessed adequate statistical power. The results shown by the TSA would indicate the presence of a spurious effect for a follow-up period of 2 years. The cause of this is unclear, but it could depend on the type of endodontic sealer or closure used for the four studies included in this meta-analysis and reported in Figure 3 [44].

The second point is the bias between studies. The bias was addressed and evaluated through the analysis of the heterogeneity indices (Higgins index and Chi^2^) and the analysis of the overlap of the confidence intervals, and in order to reduce the heterogeneity, the data meta-analyzed were collected according to the follow-up period, in order to have data that could be as homogeneous and comparable as possible. In addition, an analysis of the sources of heterogeneity through funnel plots was also performed (Figure 6) [43].

The execution of a subgroup analysis was also considered in relation to the type of endodontic sealer used but it has not been performed for too few studies within each single meta-analysis [45].

The third point is the risk of bias within the studies, and it was addressed through the evaluation of a scale, whose checklist is used and widely described in the results Section 3.3. In addition, additional evaluations of the quality of the results through the grade and the evaluation of the statistical power of the results through the TSA have been adopted in this meta-analysis.

Based on the results of the systematic review and the data from the TSA, some objectives are recommended on which retreatment and surgical endodontic retreatments studies should focus.

Primarily, to increase the level of scientific evidence, studies of good quality with a low risk of bias and which foresee the identification of univocal prognostic predictors of failure or success must be put in place. Secondly, the data from the studies could be compared much more easily if they were represented in a standardized way through the use of a format or tool that clearly presents success and failure criteria; thirdly, the data from the TSA indicate that further studies on surgical endodontic retreatments are needed compared to non-surgical endodontic retreatments with long-term outcomes.

## 5. Conclusions

On the basis of this study, surgical endodontic retreatments after 2 years of follow-up represent a predictable treatment choice with guarantee of initial successful outcome with a lower risk of failure compared to non-surgical endodontic retreatments. However, future long-term clinical studies on surgical endodontic retreatments versus non-surgical endodontic retreatments, which consider longer observation periods, are needed.

## Figures and Tables

**Figure 1 medicina-58-00894-f001:**
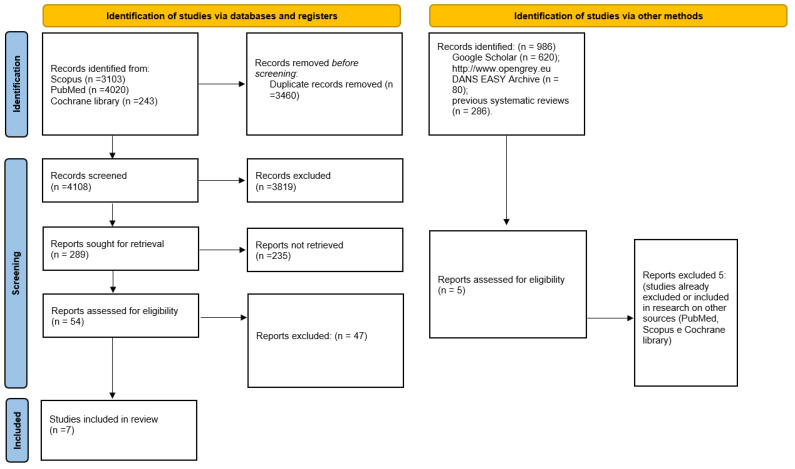
Entire selection and screening procedures as described in the PRISMA flowchart.

**Figure 2 medicina-58-00894-f002:**
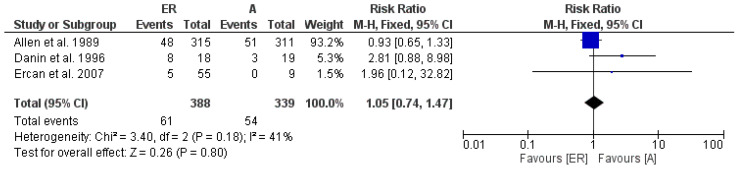
Forest plot of the fixed effects model of the meta-analysis; RR = 1.05, 95% CI: [0.74,1.47]; df = degrees of freedom; *I*^2^ = Higgins heterogeneity index, *I*^2^ < 50%, heterogeneity irrelevant; *I*^2^ > 75%, significant heterogeneity; C.I. = confidence intervals; *p* = *p* value; ER = non-surgical endodontic retreatment; A = surgical endodontic retreatments (apicoectomy). The graph for each study shows the first author and the date of publication, Risk Ratio with confidence intervals, the number of failures for each type of retreatment, the number of non-surgical and surgical endodontic retreatments and the weight of each study expressed as a percentage. The final value was expressed in bold with the relative confidence intervals, the blue squares in the center of the confidence intervals represent the effect of the study in the meta-analysis and its size represents the weight of the study. The black line shows the position of the average value, and the rhombus in light black shows the measure of the average effect.

**Figure 3 medicina-58-00894-f003:**
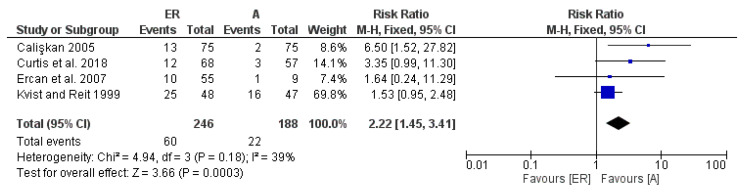
Forest plot of the fixed-effects model of the meta-analysis of the failure data within the 2-year follow-up. RR 2.22 [1.45, 3.41].

**Figure 4 medicina-58-00894-f004:**
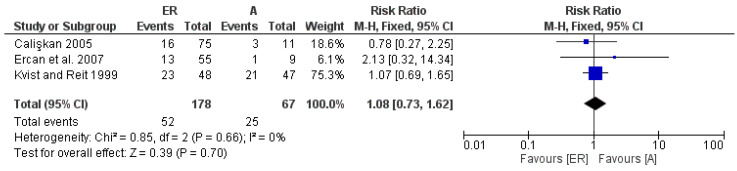
Forest plot of the fixed-effects model of the meta-analysis of the failure data within the 3–4-year follow-up. RR 1.08 [ 0.73 1.62] Chi^2^ = 0.85, df = 2 (*p* = 0.66).

**Figure 5 medicina-58-00894-f005:**
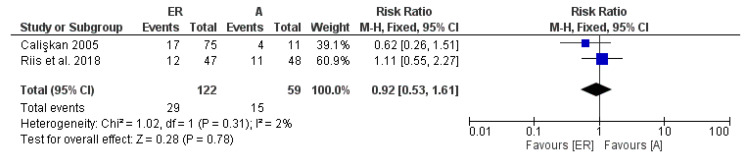
Forest plot of the fixed-effects model of the meta-analysis of the failure data within the 8–10-year follow-up. RR 0.92 [ 0.53, 1.61] Chi^2^ = 0.85, df = 2 (*p* = 0.66).

**Figure 6 medicina-58-00894-f006:**
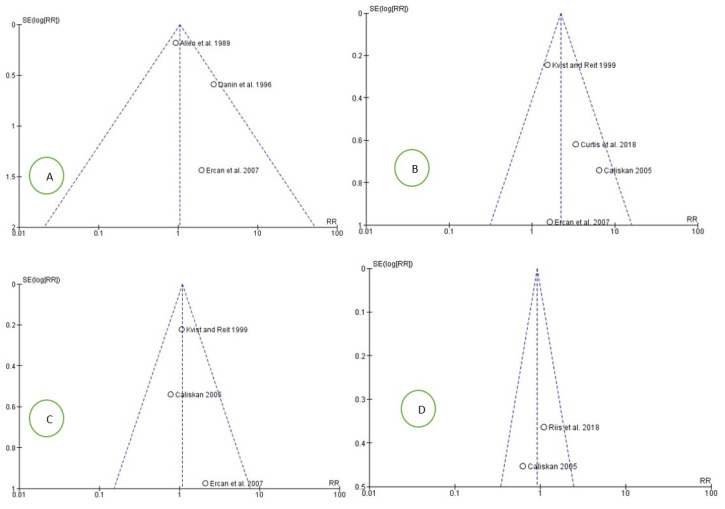
Funnel plot: (**A**) follow-up 1 year; (**B**) follow-up 2 years; (**C**) follow-up 3–4 years; (**D**) Follow-up 8–10 years. RR risk rate; SE standard error. Graphically, there are no sources of heterogeneity.

**Figure 7 medicina-58-00894-f007:**
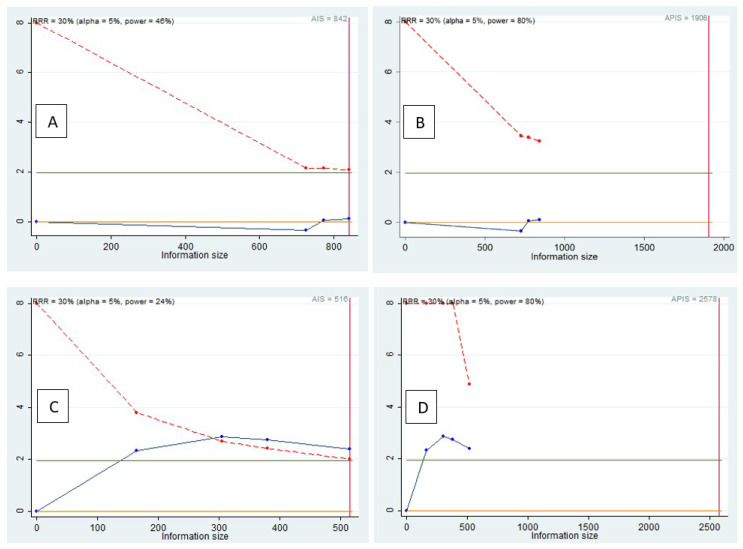
(**A**,**B**) Follow-up 1 year; (**C**,**D**) follow-up 2 year; light green line (Z = 1.98), dashed red line (monitoring boundary), blue line (cumulative z curve), red line (sample size). AIS (accrued information size (**A**,**C**)); APIS (a priori information size (**B**,**D**)).

**Table 1 medicina-58-00894-t001:** Complete overview of the search methodology.

Databases	K Words, Search Details	Records
PubMed	Search: surgical endodontic retreatment Sort by: Most Recent (“surgical procedures, operative” [MeSH Terms] OR (“surgical” [All Fields] AND “procedures” [All Fields] AND “operative” [All Fields]) OR “operative surgical procedures” [All Fields] OR “surgical” [All Fields] OR “surgically” [All Fields] OR “surgicals” [All Fields]) AND (“endodontal” [All Fields] OR “endodontic” [All Fields] OR “endodontical” [All Fields] OR “endodontically” [All Fields] OR “endodontics” [MeSH Terms] OR “endodontics” [All Fields]) AND (“retreat” [All Fields] OR “retreated” [All Fields] OR “retreating” [All Fields] OR “retreatment” [MeSH Terms] OR “retreatment” [All Fields] OR “retreatments” [All Fields] OR “retreats” [All Fields]) Translations surgical: “surgical procedures, operative” [MeSH Terms] OR (“surgical” [All Fields] AND “procedures” [All Fields] AND “operative” [All Fields]) OR “operative surgical procedures” [All Fields] OR “surgical” [All Fields] OR “surgically” [All Fields] OR “surgicals” [All Fields] endodontic: “endodontal” [All Fields] OR “endodontic” [All Fields] OR “endodontical” [All Fields] OR “endodontically” [All Fields] OR “endodontics” [MeSH Terms] OR “endodontics” [All Fields] retreatment: “retreat” [All Fields] OR “retreated” [All Fields] OR “retreating” [All Fields] OR “retreatment” [MeSH Terms] OR “retreatment” [All Fields] OR “retreatments” [All Fields] OR “retreats” [All Fields]	567
	Search: endodontic retreatment Sort by: Most Recent (“endodontal” [All Fields] OR “endodontic” [All Fields] OR “endodontical” [All Fields] OR “endodontically” [All Fields] OR “endodontics” [MeSH Terms] OR “endodontics” [All Fields]) AND (“retreat” [All Fields] OR “retreated” [All Fields] OR “retreating” [All Fields] OR “retreatment” [MeSH Terms] OR “retreatment” [All Fields] OR “retreatments” [All Fields] OR “retreats” [All Fields]) Translations endodontic: “endodontal” [All Fields] OR “endodontic” [All Fields] OR “endodontical” [All Fields] OR “endodontically” [All Fields] OR “endodontics” [MeSH Terms] OR “endodontics” [All Fields] retreatment: “retreat” [All Fields] OR “retreated” [All Fields] OR “retreating” [All Fields] OR “retreatment” [MeSH Terms] OR “retreatment” [All Fields] OR “retreatments” [All Fields] OR “retreats” [All Fields]	1698
	Search: apicoectomy Sort by: Most Recent “apicoectomy” [MeSH Terms] OR “apicoectomy” [All Fields] OR “apicoectomies” [All Fields] Translations apicoectomy: “apicoectomy” [MeSH Terms] OR “apicoectomy” [All Fields] OR “apicoectomies” [All Fields]	1755
SCOPUS	TITLE-ABS-KEY (surgical AND endodontic AND retreatment)	293
	TITLE-ABS-KEY ((endodontic AND retreatment) OR apicoectomy)	3012
Cochrane library	TITLE-ABS-KEY ((endodontic AND retreatment) OR apicoectomy)	214
	(Surgical AND endodontic AND retreatment)	29
Total	7568	

**Table 2 medicina-58-00894-t002:** Complete list of studies included in the meta-analysis.

		Summary of Data from Studies Treating Non-Surgical Endodontic Retreatment and Surgical Endodontic Retreatment [18,19,29,30,31,32,33]																	
First Author, Date	Type of Study, Country	Type of Evaluation	N. Teeth, (Patients) *	Filling Material **					Failures to Follow-Up (Year)						Failure Reason ***				
				GP	MTA	GC	P	O	1y	2y	3y	4y	8y	1y	A	B	C	D	E
Danin et al. (1996) [18]	Randomized Clinical Trial	Clinically, radiographically	19 R	19 (+RC)					8/18						8				
	Sweden		19 TC			19			3/19						3				
Curtis et al. (2018) [29]	retrospective	CBCT	68 R	68						12/68					12				
	USA		57 TC		57					3/57					3				
Calişkan (2005) [30]	retrospective	radiographically	79 R	79						13/75 ^1^		16/75	17/75		17				
	Turkey		11 TC	6			5			2/11		3/11	4/11		4				
Kvist and Reit (1999) [19]	Randomized Clinical Trial	clinically and radiographically	48 R	48						25/48		27/48 ^2^			4				2
	Sweden		47 TC	47						16/47		21/47			8				2
Allen et al. (1989) [31]	retrospective	radiographically	596 R	596					48/315										
	USA		695 TC						51/311										
Riis et al. (2018) [32]	Randomized Clinical Trial	clinically and radiographically	60 TC	60										12/47	7			2	3
	Sweden;		64 R	64 (+RC)										11/48				6	5
Ercan et al. (2007) [33]	Prospective	clinically and radiographically	59 R	59					5/55	10/55	13/55				13				
	Turkey		11 TC	11					0/9	1/9	1/9				1				

* R: non-surgically endodontic retreated, TC: surgical endodontic retreatment, (y = year). ** GP: gutta-percha, GC: glass ionomer cement, P: paste, RC: resin chloroform, O: other. *** A = endodontic problem (including infectious, e.g., periapical lesion), B = periodontal problem, C = prosthetic problem, D = tooth or root fracture, E = unknown, other or generic non-endodontic reason. (^1^) Four teeth were excluded because they were periodontally compromised. (^2^) The data were extracted from the Figure 1 graph of the study by Kvist and Reit.

**Table 3 medicina-58-00894-t003:** Assessment of risk of bias within the studies.

	Selection		Outcome	Loss to Follow-Up	Analysis	Score
Reference	Non-Response Rate	Representativeness of Sample to Target Population	Validity and Reliability of Outcome Measurement	Amount of Loss to Follow-Up	Appropriate Statistical Tests	
Danin et al. (1996) [18]	3	4	5	3	3	18
Curtis et al. (2018) [29]	3	4	4	4	4	19
Calişkan (2005) [30]	4	4	4	3	4	19
Kvist and Reit (1999) [19]	3	4	5	4	4	20
Allen et al. (1989) [31]	3	5	4	3	3	18
Riis et al. (2018) [32]	5	5	5	5	5	25
Ercan et al. (2007) [33]	4	4	4	4	4	20

**Table 4 medicina-58-00894-t004:** Evaluation of GRADE pro GDT; ⨁◯◯◯ Very low, ⨁⨁◯◯ Low, ⨁⨁⨁◯ Moderate, ⨁⨁⨁⨁ High.

Certainty Assessment							No of Patients		Effect		Certainty
No of Studies	Study Design	Risk of Bias	Inconsistency	Indirectness	Imprecision	Other Considerations	Endodontic Retreatment	Apicoectomy	Relative (95% CI)	Absolute (95% CI)	
**ER vs. A 1-year follow-up**											
3	observational studies	not serious	not serious	not serious	not serious	none	61/388 (15.7%)	54/339 (15.9%)	RR 1.05 (0.74 to 1.47)	8 more per 1.000 (from 41 fewer to 75 more)	⨁⨁◯◯ Low
**ER vs. A 2-years follow-up**											
4	observational studies	not serious	not serious	not serious	not serious	strong association	60/246 (24.4%)	22/188 (11.7%)	RR 2.22 (1.45 to 3.41)	143 more per 1.000 (from 53 more to 282 more)	⨁⨁⨁◯ Moderate
**ER vs. A 3–4-years follow-up**											
3	observational studies	not serious	not serious	not serious	not serious	none	52/178 (29.2%)	25/67 (37.3%)	RR 1.08 (0.73 to 1.62)	30 more per 1.000 (from 101 fewer to 231 more)	⨁⨁◯◯ Low
**ER vs. A 8–10-years follow-up**											
2	observational studies	not serious	not serious	not serious	not serious	none	29/122 (23.8%)	15/59 (25.4%)	RR 0.92 (0.53 to 1.61)	20 fewer per 1.000 (from 119 fewer to 155 more)	⨁⨁◯◯ Low

## Data Availability

Not applicable.

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
