# Peer review of "Comparison of Endodontic Failures between Nonsurgical Retreatment and Endodontic Surgery: Systematic Review and Meta-Analysis with Trial Sequential Analysis"

_medicina, 2022, doi:10.3390/medicina58070894_

Round 1

Reviewer 1 Report

Reviewer’s comments: Medicina

Comparison of Endodontic Failures Between Nonsurgical Retreatment and Endodontic surgery: Systematic Review and Meta‐Analysis with Trial Sequential Analysis

In this systematic review, the authors provide an analysis between the failures related to non-surgical retreatment and endodontic surgery after the advent of new endodontic biomaterials. This is an up-to-date and interesting topic; since previous reviews were published when these new technologies had not yet been incorporated into endodontic practices such as the use of ultrasound, microscopy and biomaterial. However, all these new technologies were not included in the inclusion/exclusion criteria. I recommend that the authors evaluate this point and include this information. It is extremely important to show information that is different from what has been previously published.

Author Response

Reviewer 1

Comparison of Endodontic Failures Between Nonsurgical Retreatment and Endodontic surgery: Systematic Review and Meta‐Analysis with Trial Sequential Analysis

In this systematic review, the authors provide an analysis between the failures related to non-surgical retreatment and endodontic surgery after the advent of new endodontic biomaterials. This is an up-to-date and interesting topic; since previous reviews were published when these new technologies had not yet been incorporated into endodontic practices such as the use of ultrasound, microscopy and biomaterial. However, all these new technologies were not included in the inclusion/exclusion criteria. I recommend that the authors evaluate this point and include this information. It is extremely important to show information that is different from what has been previously published.

Answer

Thank you for reviewing the manuscript.

all the advice given to me has been implemented in the manuscript and the following parts as suggested to me have been added:

  1. Exclusion criteria were not considered studies for surgical and endodontic techniques that involved the use of microscopy, ultrasonic instruments and biomaterials such as bioceramics.

  1. In this review we also focused on the type of biomaterials used to obtain the apical seal, in order to understand if it could represent a cause of a different failure rate.

According to a recent literature review conducted on root canal filling materials performed by Komabayashi et al. [40] , the Tricalcium silicate (MTA / Bioceramic) and the Salicylate (MTA fillapex, sealepex, apexit), are the categories of endodontic sealer with the major bioactive components. In fact, studies on the cytotoxicity of MTA re-vealed biomineralization and osteo-inductivity capabilities, with the additional data that, the cytocompatibility characteristics, were clearly superior to other endodontic sealers [41].

The limitations of this systematic review are that the results, even if in part sig-nificant, show a small number of participants included, as also highlighted by the TSA.

The results of this study indicate that clinicians can choose either non-surgical re-treatment or root-end surgery after failed primary root canal therapy. Finally, only in the short-term period (2 years), with balanced RR values, data regarding surgical en-dodontic retreatments, seem more in favor in terms of healing.

As shown in the results (Table 2) Curtis et al. [29], perform root-end fillings for surgical endodontic retreatments with gray or white ProRoot MTA (Dentsply, Tulsa, OK), or EndoSequence BC Root Repair Material (Brassler USA, Savannah, GA), while for retreatments, an endodontic sealear based zinc oxide eugenol (Roth 811 -Roth in-ternational, Chicago, IL, USA) was used.

Ercam et al. [33], performed root canal closure through the use of gutta-percha points and a Salicylate Ca (OH) 2 (Sealapex; Sybron / Kerr, Romulus, MI, USA): they reported 14 endodontic failures (13 in the non-surgical endodontic retreatment group and 1 in the surgical endodontic retreatment group), of which totally 10 causes were due to an error in filling the canal (underfilling, overfilling, poorly condensed filling), totally 2 due, to broken instrumentation and and finally 2 due to loss of the coronal seal.

In the other included studies, bioceramics or more generally, Tricalcium silicate or Salycilati, were not used as root-end fillings.

 Indeed, the studies by Kvist and Reit)[19], Allen et al. [31] and Riss et al. [32] proposed the use of gutta-percha, Danin et al. [18], the glass ionomer cement (Chem-fill II, De Trey, Zurich, Switzerland) for apical closure of teeth undergoing sur-gery),  while Calistan [30] performed the canal closures using a silicone type sealer (Diaket -ESPE, Seefeld, Germany),  with gutta-percha cones and calcium hydroxide paste  were used for the apical closure of the surgically treated teeth. For these stud-ies, the causes of failure were extensively described in Table 2.

The data of the failures extrapolated and presented in Table 2, do not reveal suf-ficient data to be able to perform an analysis of the subgroups, according to the type of sealer used in the course of surgical endodontic retreatment.

Best regards Mario Dioguardi

Reviewer 2 Report

This research provided an updated value of the Risk Ratio between the two types of treatment; non‐surgical-endodontic retreatment or an endodontic surgery. Some comments.

 Please remove the full stop from the title.

 Abstract

It is better to provide specific key words used in search criteria.

Line 32-33 “In addition, the trial sequency analysis (TSA), was performed for the analysis of the statistical power of the results and the GRADE for the quality of the evidence” should be moved to the method rather than in the results.

 Introduction

It is better to provide criteria for the non-surgical and surgical endodontic treatment.

Results:

Line 300-301. There is the repetition of the words “second”.

Discussion:

Please add more on Trial Sequential Analysis.

Please add limitations of this research.

Author Response

Reviewer 2

This research provided an updated value of the Risk Ratio between the two types of treatment; non‐surgical-endodontic retreatment or an endodontic surgery. Some comments.

 Please remove the full stop from the title.

 Abstract

It is better to provide specific key words used in search criteria.

Line 32-33 “In addition, the trial sequency analysis (TSA), was performed for the analysis of the statistical power of the results and the GRADE for the quality of the evidence” should be moved to the method rather than in the results.

 Introduction

It is better to provide criteria for the non-surgical and surgical endodontic treatment.

Results:

Line 300-301. There is the repetition of the words “second”.

Discussion:

Please add more on Trial Sequential Analysis.

Please add limitations of this research.

Answer

Thank you for reviewing the manuscript.

all the advice given to me has been implemented in the manuscript and the following parts as suggested to me have been added:

  1. Title: removed the Full stop at the end of the title.
  2. Abstract: More specific keywords have been added as suggested (Keywords: Endodontic, Endodontic retreatment; Apicoectomy; Surgical endodontic retreatment; Trial Sequential Analysis; Endodontic failure, Endodontically, Endodontic surgery.); The following sentence has been moved to the materials e methods “In addition, the trial sequency analysis (TSA), was performed for the analysis of the statistical power of the results and the GRADE for the quality of the evidence”;
  3. Introduction: as suggested the following criteria have been implemented in the text: In accordance with Karabucak and Setzer [1], the criteria which should guide the clinician in choosing between performing a non-surgical endodontic retreatment and a surgical endodontic retreatment, concern the following points:
  4. Evaluation of the coronal seal: the presence of an adequate coronal restoration or a still functional prosthetic crown that seals the endodontium coronally, or whose removal involves the non-restorability, can lead to the choice towards surgical endodontic retreatment, on the contrary a restoration no longer adequate, with good access to the endodontium, may lead to the choice of non-surgical endodontic retreatment[2];
  5. Radiographic evaluation of root canal obturation: the presence of untreated canals, and a coarse filling of the canals, may lead to the choice of non-surgical endodontic retreatment, on the contrary, a surgical approach could be recommended in the presence of an apparent adequate apical seal or in the presence of a root canal obturation that is far from the radiographic apex of only 1 mm (in the presence of course of symptoms of a periapical pathology), a surgical approach should be evaluated [3-5].
  • The presence of clinical complications such as: the finding of fractured instruments (apical third of the root), the presence of zipping of internal resorptions and presence of root perforations  or ledge formations; in these cases the surgical approach may be a suitable choice [6].
  1. In the presence of recurrent infections in which a root canal retreatment has already been done, a surgical endodontic retreatment is absolutely to be considered as a therapeutic choice [7].
  2. Results: the repeated word has been removed (second)
  3. Discussion: as suggested some additional information on the TSA has been added: In fact, the TSA conducted on the RR data on failures at one year, shows that for the 3 included studies there is no adequate statistical power of the results, indicating the absence of evidence, with a possibility of a false negative result; instead, if we con-sider a follow-up period of 2 years, the condition changes, and the TSA shows statisti-cally significant data, but with the possibility of incurring a false positive result for the presence of a spurious effect determined by confounding factors, indicating how the optimal number of the patients included in the studies, to be confident of a true posi-tive result, is 2578.

The majority of the systematic reviews with meta-analyzes data, including the Cochrane studies, present a number of studies with randomized participants too low to obtain adequate statistical power, that would allow an adequate evaluation of the intervention effect size. The results deriving from a meta-analysis with few studies, and consequently with few participants, could be not very credible, with the risk of overestimation or of being underestimated, as they do not possess sufficient statistical power to accept or reject interventions even with large effect sizes [38] .

  1. limit have been implemented as required: The main risk incurred by reading the resulting data of this meta-analysis is that of over-interpreting the data at 2 years: the analysis we conducted did not show statis-tically significant differences compared to 1, 3-4 and 8-10 years Suggesting one treat-ment or the other, the 2-year period could be considered suitable in many studies but, the success of an endodontic therapy should be evaluated in reasonably longer fol-low-up periods, consistent with the planned duration of an endodontic therapy.

The limitations of this systematic review are that the results, even if in part sig-nificant, show a small number of participants included, as also highlighted by the TSA.

Best regards Mario Dioguardi

  1. Karabucak, B.; Setzer, F. Criteria for the ideal treatment option for failed endodontics: surgical or nonsurgical? Compend Contin Educ Dent 2007, 28, 304-310; quiz 311, 332.
  2. Kalyani, P.; Patwa, N.; Gupta, N.; Bhatt, A.; Saha, S.; Kanjani, V. Clinical and radiographic assessment of post-treatment endodontic disease by primary healthcare professionals: A hospital-based 1-year follow-up. Journal of family medicine and primary care 2022, 11, 1114-1118, doi:10.4103/jfmpc.jfmpc_2033_20.
  3. Javed, M.Q.; AlAttas, M.H.; Bhatti, U.A.; Dutta, S.D. Retrospective audit for quality assessment of root fillings performed by undergraduate dental students in clinics. J Taibah Univ Med Sci 2022, 17, 297-303, doi:10.1016/j.jtumed.2021.10.005.
  4. Aga, N.; Thakur, M.K.; Agwan, M.A.S.; Eisa, M.; Habshi, A.Y.; Azeem, S. Evaluation of Quality of Endodontic Re-Treatment and Changes in Periapical Status. J Pharm Bioallied Sci 2021, 13, S379-s382, doi:10.4103/jpbs.JPBS_814_20.
  5. Al Shehadat, S.; El-Kishawi, M.; AlMudalal, A.; AlSaqer, A.; Nassar, A.; Zihlif, L.; Mahmoud, Y.; Nagendrababu, V.; Vinothkumar, T.S. An Audit of the Technical Quality and Iatrogenic Errors of Root Canal Treatment by Undergraduate Dental Students at the University of Sharjah. Eur J Dent 2022, 10.1055/s-0042-1743150, doi:10.1055/s-0042-1743150.
  6. Kalogeropoulos, K.; Xiropotamou, A.; Koletsi, D.; Tzanetakis, G.N. The Effect of Cone-Beam Computed Tomography (CBCT) Evaluation on Treatment Planning after Endodontic Instrument Fracture. Int J Environ Res Public Health 2022, 19, doi:10.3390/ijerph19074088.
  7. Abusrewil, S.; Alshanta, O.A.; Albashaireh, K.; Alqahtani, S.; Nile, C.J.; Scott, J.A.; McLean, W. Detection, treatment and prevention of endodontic biofilm infections: what's new in 2020? Crit Rev Microbiol 2020, 46, 194-212, doi:10.1080/1040841x.2020.1739622.

Reviewer 3 Report

Dear Authors,

I am honored to review this article entitled: "Comparison of Endodontic Failures Between Nonsurgical Re‐treatment and Endodontic surgery: Systematic Review and Meta‐Analysis with Trial Sequential Analysis".

It is clear that the authors have done a lot of revision work; however, I do not consider the study design to be adequate, since not all the variables that influence the prognosis of previous root canal treatment, root canal re-treatment and endodontic surgery have been taken into account. There is also no reference to the etiology of endodontic therapy failure. The selected articles are scarce and some are very old (1989, 1996, 1999). The regeneration techniques of endodontic surgeries have not been taken into account either.

Author Response

Reviewer 3

Dear Authors,

I am honored to review this article entitled: "Comparison of Endodontic Failures Between Nonsurgical Re‐treatment and Endodontic surgery: Systematic Review and Meta‐Analysis with Trial Sequential Analysis".

It is clear that the authors have done a lot of revision work; however, I do not consider the study design to be adequate, since not all the variables that influence the prognosis of previous root canal treatment, root canal re-treatment and endodontic surgery have been taken into account. There is also no reference to the etiology of endodontic therapy failure. The selected articles are scarce and some are very old (1989, 1996, 1999). The regeneration techniques of endodontic surgeries have not been taken into account either.

Answer

Thank you for reviewing the manuscript. Your suggestions have been extremely helpful in improving the quality of the manuscript:

in order to improve the design of the study whose protocol has been registered on PROSPERO with number of number CRD42021273533 the following considerations have been implemented:

  1. The criteria by which the clinician should guide in the choice between performing a non-surgical endodontic retreatment and a surgical endodontic retreatment concern the following points added to the introduction: In accordance with Karabucak and Setzer [1], concern the following points:
  2. Evaluation of the coronal seal: the presence of an adequate coronal restoration or a still functional prosthetic crown that seals the endodontium coronally, or whose removal involves the non-restorability, can lead to the choice towards surgical endodontic retreatment, on the contrary a restoration no longer adequate, with good access to the endodontium, may lead to the choice of non-surgical endodontic retreatment[2];
  3. Radiographic evaluation of root canal obturation: the presence of untreated canals, and a coarse filling of the canals, may lead to the choice of non-surgical endodontic retreatment, on the contrary, a surgical approach could be recommended in the presence of an apparent adequate apical seal or in the presence of a root canal obturation that is far from the radiographic apex of only 1 mm (in the presence of course of symptoms of a periapical pathology), a surgical approach should be evaluated [3-5].
  • The presence of clinical complications such as: the finding of fractured instruments (apical third of the root), the presence of zipping of internal resorptions and presence of root perforations  or ledge formations; in these cases the surgical approach may be a suitable choice [6].
  1. In the presence of recurrent infections in which a root canal retreatment has already been done, a surgical endodontic retreatment is absolutely to be considered as a therapeutic choice [7].
  2. A careful assessment of possible BIAS was performed within the systematic review: The analysis of possible biases within the present systematic review was ad-dressed and planned on 3 fundamental points. The first point is the presence of a pub-lication Bias; the real issue was given from the small number of articles included  (7 for this systematic review, table 2). The publication bias is a problem that afflicts sys-tematic reviews with meta-analyzes data, because the scientific literature tends not to publish statistically insignificant data and, an attempt has been made to minimize it through analysis of the gray literature and of the sources that also include abstracts of conferences, in order to intercept those reports of studies performed but not published [42].

Furthermore, to assess the presence of a probable publication bias, a visual analy-sis of the symmetry of the funnel plots was performed. However, was not possible to clearly evaluate the issue, due to the small number of studies[43]. Furthermore, in the light of the few included studies, TSA was performed to determine if there was an ad-equate number of participants and, if the results possessed adequate statistical power. The results shown by the TSA, would indicate the presence of a spurious effect for a follow-up period of 2 years :the cause of which is unclear but could depend on the type of endodontic sealer or closure used for the 4 studies included in this meta-analysis and reported in Figure 3)[44].

The second point is the bias between studies. The bias was addressed and evalu-ated through the analysis of the heterogeneity indices (Higgins index and Chi2), the analysis of the overlap of the confidence intervals and, in order to reduce the hetero-geneity, the data meta-analyzed, were performed according to the follow-up period, in order to have data that could be homogeneous and comparable as possible. In addition, an analysis of the sources of heterogeneity through the funnel plots was also per-formed (Figure 6)[43].

The execution of a subgroup analysis was also considered in relation to the type of endodontic sealer used but was not performed for too few studies within each single meta-analysis[45].

The third point is the risk of bias within the studies and was addressed through the evaluation of a scale, whose checklist is used and widely described in the results section (3.3). In addition, additional evaluations on the quality of the results through the grade and the evaluation of the statistical power of the results through the TSA, have been adopted in this meta-analysis.

  1. Possible variables that can influence, giving a spurious effect as evidenced in the TSA, the retreatment failure rate have been added as suggested: The main risk incurred by reading the resulting data of this meta-analysis is that of over-interpreting the data at 2 years: the analysis we conducted did not show statis-tically significant differences compared to 1, 3-4 and 8-10 years Suggesting one treat-ment or the other, the 2-year period could be considered suitable in many studies but, the success of an endodontic therapy should be evaluated in reasonably longer fol-low-up periods, consistent with the planned duration of an endodontic therapy.

Surgical endodontic retreatment has disadvantages that could nullify the im-provement  obtained in the short term. In fact Riis et al. [32] reports as causes of fail-ure for the surgical endodontic retreatment group at 10 years, 2 cases for vertical root fracture and 7 cases for non-healing. The apical seals, in this case, were obtained by using gutta-percha for both groups.

Furthermore, it is increasingly clear from the data in the literature that, among the main causes of failure in teeth with surgical endodontic retreatment, is the vertical fracture. In fact, this issue emerges from a study on the incidence of vertical root frac-tures between conventional versus surgical endodontic retreatment conducted by Karygianni et al. [39], that 62.31% of the teeth with vertical root fracture had under-gone a combination of conventional endodontic retreatment in addition to surgical endodontic retreatment. Other causes for the persistence of the lesion, could be found in the quality of the existing root canal filling, or in the solubility of the retrograde fill-ing.

In this review we also focused on the type of biomaterials used to obtain the apical seal, in order to understand if it could represent a cause of a different failure rate.

According to a recent literature review conducted on root canal filling materials performed by Komabayashi et al. [40] , the Tricalcium silicate (MTA / Bioceramic) and the Salicylate (MTA fillapex, sealepex, apexit), are the categories of endodontic sealer with the major bioactive components. In fact, studies on the cytotoxicity of MTA re-vealed biomineralization and osteo-inductivity capabilities, with the additional data that, the cytocompatibility characteristics, were clearly superior to other endodontic sealers [41].

The limitations of this systematic review are that the results, even if in part sig-nificant, show a small number of participants included, as also highlighted by the TSA.

The results of this study indicate that clinicians can choose either non-surgical re-treatment or root-end surgery after failed primary root canal therapy. Finally, only in the short-term period (2 years), with balanced RR values, data regarding surgical en-dodontic retreatments, seem more in favor in terms of healing.

As shown in the results (Table 2) Curtis et al. [29], perform root-end fillings for surgical endodontic retreatments with gray or white ProRoot MTA (Dentsply, Tulsa, OK), or EndoSequence BC Root Repair Material (Brassler USA, Savannah, GA), while for retreatments, an endodontic sealear based zinc oxide eugenol (Roth 811 -Roth in-ternational, Chicago, IL, USA) was used.

Ercam et al. [33], performed root canal closure through the use of gutta-percha points and a Salicylate Ca (OH) 2 (Sealapex; Sybron / Kerr, Romulus, MI, USA): they reported 14 endodontic failures (13 in the non-surgical endodontic retreatment group and 1 in the surgical endodontic retreatment group), of which totally 10 causes were due to an error in filling the canal (underfilling, overfilling, poorly condensed filling), totally 2 due, to broken instrumentation and and finally 2 due to loss of the coronal seal.

In the other included studies, bioceramics or more generally, Tricalcium silicate or Salycilati, were not used as root-end fillings.

  Indeed, the studies by Kvist and Reit)[19], Allen et al. [31] and Riss et al. [32] proposed the use of gutta-percha, Danin et al. [18], the glass ionomer cement (Chem-fill II, De Trey, Zurich, Switzerland) for apical closure of teeth undergoing sur-gery),  while Calistan [30] performed the canal closures using a silicone type sealer (Diaket -ESPE, Seefeld, Germany),  with gutta-percha cones and calcium hydroxide paste  were used for the apical closure of the surgically treated teeth. For these stud-ies, the causes of failure were extensively described in Table 2.

  The data of the failures extrapolated and presented in Table 2, do not reveal suf-ficient data to be able to perform an analysis of the subgroups, according to the type of sealer used in the course of surgical endodontic retreatment.

  1. 7 articles are included in the systematic review Danin et al. (1996)[8], Curtis et al. (2018)[9], Calişkan (2005)[10], Kvist and Reit (1999)[11], Allen et al. (1989)[12], Riis et al. (2018)[13], and Ercan et al. (2007)[14]. of which the other 4 cover a period from 2005 to 2018,
  2. Taking into account endodontic regeneration techniques was too off topic for a systematic review with meta-analysis on the confrontation between endodontic surgery and non-surgical retreatments; but surely it can be a starting point for the execution of a mapping review or on a more specific systematic review on endodontic regeneration once a precise outcome has been identified.

Best regards Mario Dioguardi

  1. Karabucak, B.; Setzer, F. Criteria for the ideal treatment option for failed endodontics: surgical or nonsurgical? Compend Contin Educ Dent 2007, 28, 304-310; quiz 311, 332.
  2. Kalyani, P.; Patwa, N.; Gupta, N.; Bhatt, A.; Saha, S.; Kanjani, V. Clinical and radiographic assessment of post-treatment endodontic disease by primary healthcare professionals: A hospital-based 1-year follow-up. Journal of family medicine and primary care 2022, 11, 1114-1118, doi:10.4103/jfmpc.jfmpc_2033_20.
  3. Javed, M.Q.; AlAttas, M.H.; Bhatti, U.A.; Dutta, S.D. Retrospective audit for quality assessment of root fillings performed by undergraduate dental students in clinics. J Taibah Univ Med Sci 2022, 17, 297-303, doi:10.1016/j.jtumed.2021.10.005.
  4. Aga, N.; Thakur, M.K.; Agwan, M.A.S.; Eisa, M.; Habshi, A.Y.; Azeem, S. Evaluation of Quality of Endodontic Re-Treatment and Changes in Periapical Status. J Pharm Bioallied Sci 2021, 13, S379-s382, doi:10.4103/jpbs.JPBS_814_20.
  5. Al Shehadat, S.; El-Kishawi, M.; AlMudalal, A.; AlSaqer, A.; Nassar, A.; Zihlif, L.; Mahmoud, Y.; Nagendrababu, V.; Vinothkumar, T.S. An Audit of the Technical Quality and Iatrogenic Errors of Root Canal Treatment by Undergraduate Dental Students at the University of Sharjah. Eur J Dent 2022, 10.1055/s-0042-1743150, doi:10.1055/s-0042-1743150.
  6. Kalogeropoulos, K.; Xiropotamou, A.; Koletsi, D.; Tzanetakis, G.N. The Effect of Cone-Beam Computed Tomography (CBCT) Evaluation on Treatment Planning after Endodontic Instrument Fracture. Int J Environ Res Public Health 2022, 19, doi:10.3390/ijerph19074088.
  7. Abusrewil, S.; Alshanta, O.A.; Albashaireh, K.; Alqahtani, S.; Nile, C.J.; Scott, J.A.; McLean, W. Detection, treatment and prevention of endodontic biofilm infections: what's new in 2020? Crit Rev Microbiol 2020, 46, 194-212, doi:10.1080/1040841x.2020.1739622.
  8. Danin, J.; Strömberg, T.; Forsgren, H.; Linder, L.E.; Ramsköld, L.O. Clinical management of nonhealing periradicular pathosis. Surgery versus endodontic retreatment. Oral Surg Oral Med Oral Pathol Oral Radiol Endod 1996, 82, 213-217, doi:10.1016/s1079-2104(96)80259-9.
  9. Curtis, D.M.; VanderWeele, R.A.; Ray, J.J.; Wealleans, J.A. Clinician-centered Outcomes Assessment of Retreatment and Endodontic Microsurgery Using Cone-beam Computed Tomographic Volumetric Analysis. J Endod 2018, 44, 1251-1256, doi:10.1016/j.joen.2018.03.016.
  10. Calişkan, M.K. Nonsurgical retreatment of teeth with periapical lesions previously managed by either endodontic or surgical intervention. Oral Surg Oral Med Oral Pathol Oral Radiol Endod 2005, 100, 242-248, doi:10.1016/j.tripleo.2004.09.014.
  11. Kvist, T.; Reit, C. Results of endodontic retreatment: a randomized clinical study comparing surgical and nonsurgical procedures. J Endod 1999, 25, 814-817, doi:10.1016/s0099-2399(99)80304-5.
  12. Allen, R.K.; Newton, C.W.; Brown, C.E., Jr. A statistical analysis of surgical and nonsurgical endodontic retreatment cases. J Endod 1989, 15, 261-266, doi:10.1016/s0099-2399(89)80221-3.
  13. Riis, A.; Taschieri, S.; Del Fabbro, M.; Kvist, T. Tooth Survival after Surgical or Nonsurgical Endodontic Retreatment: Long-term Follow-up of a Randomized Clinical Trial. J Endod 2018, 44, 1480-1486, doi:10.1016/j.joen.2018.06.019.
  14. Ercan, E.; Dalli, M.; Duülgergil, C.T.; Yaman, F. Effect of intracanal medication with calcium hydroxide and 1% chlorhexidine in endodontic retreatment cases with periapical lesions: an in vivo study. J Formos Med Assoc 2007, 106, 217-224, doi:10.1016/s0929-6646(09)60243-6.

Reviewer 4 Report

The present study is a very interesting work comparing the success rate of surgical and non-surgical endodontic treatments. The authors have performed a systematic review of relevant literature and a meta-analysis with trial sequential analysis. With regard to the results obtained, there was an advantage of surgical intervention compared to purely orthograde endodontic revision after 2 years. 

Regarding the results, however, it should be urgently noted not to overinterpret those found here. Looking at the comparisons after 1, 3-4, and 8-10 years, no statistically significant differences in favor of one or the other therapy could be detected in the literature. Although other studies consider the 2-year period to be reasonable for grouping therapy into successful and unsuccessful, it should not be ignored that an observation period should also be viewed in terms of long-term success. In this respect, a limitation to 2 years, even if partially supported by literature, represents a selective view also of the planned survival duration of the implemented endodontic therapy. It should be noted that endodontic surgery must also include factors which ultimately cancel out this very advantage for the surgical intervention in the case of a longer observation period (development of retrograde defects in the course of the surgical intervention, change in the crown-root ratio due to the retro surgery, solubility of the retrograde filling, quality of the existing root canal filling, etc.). This aspect should be clearly elaborated in the statement. However, if longer observation periods are not available in the literature to allow a comprehensive statement regarding the comparison of these two forms of therapy, this should be clearly mentioned. In any case, the conclusion in the abstract is not meaningful in its current form with regard to a general statement, even though a longer observation period was recommended in the conclusion as chapter 5. Based on the current statement in the conclusion of the abstract, the reader may get the impression that surgical intervention is generally superior to orthograde retreatment.

Also, the introduction should be specifically included in the results section, discussion and conclusion. New studies and new bioactive filling materials should be considered here in this study. With regard to the availability of bioactive materials, these have also been available as root canal sealers for more than 10 years. Considering the fact that some bioactive filling materials are also used in retrograde surgery, the objective of the study naturally raises the question of the extent to which these bioactive sealers were also taken into account in the context of orthograde retreatment.

The sources are still incorrectly formatted. some of the journals are abbreviated, some are written out in full. Partly upper case and partly lower case have been used. Please modify according to Medicina's guidelines.

Thank you.

Author Response

Reviewer 4

The present study is a very interesting work comparing the success rate of surgical and non-surgical endodontic treatments. The authors have performed a systematic review of relevant literature and a meta-analysis with trial sequential analysis. With regard to the results obtained, there was an advantage of surgical intervention compared to purely orthograde endodontic revision after 2 years. 

Regarding the results, however, it should be urgently noted not to overinterpret those found here. Looking at the comparisons after 1, 3-4, and 8-10 years, no statistically significant differences in favor of one or the other therapy could be detected in the literature. Although other studies consider the 2-year period to be reasonable for grouping therapy into successful and unsuccessful, it should not be ignored that an observation period should also be viewed in terms of long-term success. In this respect, a limitation to 2 years, even if partially supported by literature, represents a selective view also of the planned survival duration of the implemented endodontic therapy. It should be noted that endodontic surgery must also include factors which ultimately cancel out this very advantage for the surgical intervention in the case of a longer observation period (development of retrograde defects in the course of the surgical intervention, change in the crown-root ratio due to the retro surgery, solubility of the retrograde filling, quality of the existing root canal filling, etc.). This aspect should be clearly elaborated in the statement. However, if longer observation periods are not available in the literature to allow a comprehensive statement regarding the comparison of these two forms of therapy, this should be clearly mentioned. In any case, the conclusion in the abstract is not meaningful in its current form with regard to a general statement, even though a longer observation period was recommended in the conclusion as chapter 5. Based on the current statement in the conclusion of the abstract, the reader may get the impression that surgical intervention is generally superior to orthograde retreatment.

Also, the introduction should be specifically included in the results section, discussion and conclusion. New studies and new bioactive filling materials should be considered here in this study. With regard to the availability of bioactive materials, these have also been available as root canal sealers for more than 10 years. Considering the fact that some bioactive filling materials are also used in retrograde surgery, the objective of the study naturally raises the question of the extent to which these bioactive sealers were also taken into account in the context of orthograde retreatment.

The sources are still incorrectly formatted. some of the journals are abbreviated, some are written out in full. Partly upper case and partly lower case have been used. Please modify according to Medicina's guidelines.

Thank you.

Answer

Thank you for the suggestions and comments that have been helpful in improving the manuscript: below are the answers and changes to your comments.

Regarding the results, however, it should be urgently noted not to overinterpret those found here. Looking at the comparisons after 1, 3-4, and 8-10 years, no statistically significant differences in favor of one or the other therapy could be detected in the literature. Although other studies consider the 2-year period to be reasonable for grouping therapy into successful and unsuccessful, it should not be ignored that an observation period should also be viewed in terms of long-term success. In this respect, a limitation to 2 years, even if partially supported by literature, represents a selective view also of the planned survival duration of the implemented endodontic therapy. It should be noted that endodontic surgery must also include factors which ultimately cancel out this very advantage for the surgical intervention in the case of a longer observation period (development of retrograde defects in the course of the surgical intervention, change in the crown-root ratio due to the retro surgery, solubility of the retrograde filling, quality of the existing root canal filling, etc.). This aspect should be clearly elaborated in the statement. However, if longer observation periods are not available in the literature to allow a comprehensive statement regarding the comparison of these two forms of therapy, this should be clearly mentioned. In any case, the conclusion in the abstract is not meaningful in its current form with regard to a general statement, even though a longer observation period was recommended in the conclusion as chapter 5. Based on the current statement in the conclusion of the abstract, the reader may get the impression that surgical intervention is generally superior to orthograde retreatment.

  1. The following parts have been added in response following the directions of your comments;

The main risk incurred by reading the resulting data of this meta-analysis is that of over-interpreting the data at 2 years: the analysis we conducted did not show statistically significant differences compared to 1, 3-4 and 8-10 years Suggesting one treatment or the other, the 2-year period could be considered suitable in many studies but, the success of an endodontic therapy should be evaluated in reasonably longer follow-up periods, consistent with the planned duration of an endodontic therapy.

Surgical endodontic retreatment has disadvantages that could nullify the improvement  obtained in the short term. In fact Riis et al. [32] reports as causes of failure for the surgical endodontic retreatment group at 10 years, 2 cases for vertical root fracture and 7 cases for non-healing. The apical seals, in this case, were obtained by using gutta-percha for both groups.

Furthermore, it is increasingly clear from the data in the literature that, among the main causes of failure in teeth with surgical endodontic retreatment, is the vertical fracture. In fact, this issue emerges from a study on the incidence of vertical root fractures between conventional versus surgical endodontic retreatment conducted by Karygianni et al. [39], that 62.31% of the teeth with vertical root fracture had undergone a combination of conventional endodontic retreatment in addition to surgical endodontic retreatment. Other causes for the persistence of the lesion, could be found in the quality of the existing root canal filling, or in the solubility of the retrograde filling.

In this review we also focused on the type of biomaterials used to obtain the apical seal, in order to understand if it could represent a cause of a different failure rate.

According to a recent literature review conducted on root canal filling materials performed by Komabayashi et al. [40] , the Tricalcium silicate (MTA / Bioceramic) and the Salicylate (MTA fillapex, sealepex, apexit), are the categories of endodontic sealer with the major bioactive components. In fact, studies on the cytotoxicity of MTA revealed biomineralization and osteo-inductivity capabilities, with the additional data that, the cytocompatibility characteristics, were clearly superior to other endodontic sealers [41]

  1. furthermore the conclusions in the abstract have been modified as follows:

The results of the present meta-analysis, show that in the long term, the risk of failure is identical for the 2 groups, there is only a slightly higher risk of failure for non-surgical endodontic retreatments, when only 2 years of follow-up are considered..

Also, the introduction should be specifically included in the results section, discussion and conclusion. New studies and new bioactive filling materials should be considered here in this study. With regard to the availability of bioactive materials, these have also been available as root canal sealers for more than 10 years. Considering the fact that some bioactive filling materials are also used in retrograde surgery, the objective of the study naturally raises the question of the extent to which these bioactive sealers were also taken into account in the context of orthograde retreatment.

The following parts have been added in response following the directions of your comments:

In this review we also focused on the type of biomaterials used to obtain the apical seal, in order to understand if it could represent a cause of a different failure rate.

According to a recent literature review conducted on root canal filling materials performed by Komabayashi et al. [40] , the Tricalcium silicate (MTA / Bioceramic) and the Salicylate (MTA fillapex, sealepex, apexit), are the categories of endodontic sealer with the major bioactive components. In fact, studies on the cytotoxicity of MTA revealed biomineralization and osteo-inductivity capabilities, with the additional data that, the cytocompatibility characteristics, were clearly superior to other endodontic sealers [41].

The limitations of this systematic review are that the results, even if in part significant, show a small number of participants included, as also highlighted by the TSA.

The results of this study indicate that clinicians can choose either non-surgical retreatment or root-end surgery after failed primary root canal therapy. Finally, only in the short-term period (2 years), with balanced RR values, data regarding surgical endodontic retreatments, seem more in favor in terms of healing.

As shown in the results (Table 2) Curtis et al. [29], perform root-end fillings for surgical endodontic retreatments with gray or white ProRoot MTA (Dentsply, Tulsa, OK), or EndoSequence BC Root Repair Material (Brassler USA, Savannah, GA), while for retreatments, an endodontic sealear based zinc oxide eugenol (Roth 811 -Roth international, Chicago, IL, USA) was used.

Ercam et al. [33], performed root canal closure through the use of gutta-percha points and a Salicylate Ca (OH) 2 (Sealapex; Sybron / Kerr, Romulus, MI, USA): they reported 14 endodontic failures (13 in the non-surgical endodontic retreatment group and 1 in the surgical endodontic retreatment group), of which totally 10 causes were due to an error in filling the canal (underfilling, overfilling, poorly condensed filling), totally 2 due, to broken instrumentation and and finally 2 due to loss of the coronal seal.

In the other included studies, bioceramics or more generally, Tricalcium silicate or Salycilati, were not used as root-end fillings.

  Indeed, the studies by Kvist and Reit)[19], Allen et al. [31] and Riss et al. [32] proposed the use of gutta-percha, Danin et al. [18], the glass ionomer cement (Chem-fill II, De Trey, Zurich, Switzerland) for apical closure of teeth undergoing surgery),  while Calistan [30] performed the canal closures using a silicone type sealer (Diaket -ESPE, Seefeld, Germany),  with gutta-percha cones and calcium hydroxide paste  were used for the apical closure of the surgically treated teeth. For these studies, the causes of failure were extensively described in Table 2.

  The data of the failures extrapolated and presented in Table 2, do not reveal sufficient data to be able to perform an analysis of the subgroups, according to the type of sealer used in the course of surgical endodontic retreatment.

The sources are still incorrectly formatted. some of the journals are abbreviated, some are written out in full. Partly upper case and partly lower case have been used. Please modify according to Medicina's guidelines.

References were formatted according to MDPI guidelines and Medicina's guidelines.

Best Regards Mario Dioguardi

Reviewer 5 Report

Dear Authors,

Your article is a very complex meta-analysis of the success rate of endodontic orthograde  retreatment in comparison to apical surgery in cases of endodontic failures, at various periods of time (which is still a debate in Endodontics).

Unfortunately, only a low number of studies meets all the inclusion/exclusion criteria, to be used as data base, as you conclude, but your analysis is relevant to the topic.

The Introduction section, Materials and Methods, Results, are very well written, the analysis are very complex and well conducted.

The Discussion section is well sustained by literature data.

I recommend publication after minor revision.

Author Response

Dear Authors,

Your article is a very complex meta-analysis of the success rate of endodontic orthograde  retreatment in comparison to apical surgery in cases of endodontic failures, at various periods of time (which is still a debate in Endodontics).

Unfortunately, only a low number of studies meets all the inclusion/exclusion criteria, to be used as data base, as you conclude, but your analysis is relevant to the topic.

The Introduction section, Materials and Methods, Results, are very well written, the analysis are very complex and well conducted.

The Discussion section is well sustained by literature data.

I recommend publication after minor revision.

ANSWER

Thanks for reviewing the article and for the comments. all the changes made to the manuscript have been traced on the file in word.

Best regards Mario Dioguardi
